# Herbivore community function shapes resilience and bistability of coral reefs

Shayna A. Sura¤*, James O. Lloyd-Smith‡, Peggy Fong‡

Department of Ecology and Evolutionary Biology, University of California Los Angeles, Los Angeles, California, United States of America

‡ JOL-S and PF are Joint Senior Authors.
¤ Current Address: National Center for Ecological Analysis and Synthesis, University of California Santa Barbara, Santa Barbara, California, USA
* ssura@ucla.edu, sura@nceas.ucsb.edu

## Abstract

Ecological communities globally are shifting to degraded states, motivating research into attributes supporting resilience or leading to alternative stable states. Coral reef communities are particularly vulnerable as they are facing myriad anthropogenic impacts that contribute to shifts away from coral dominance, motivating much research on whether these shifts are gradual and reversible transitions versus alternative stable states. Empirical studies demonstrate functionally-diverse herbivorous fish communities support coral reef resilience to anthropogenic impacts. However, few coral reef models incorporate herbivore and algal functional groups and quantify their effects on reef resilience and alternative stable states. We built a coral reef model that includes herbivorous fish functional groups and their algal targets and explored how this expansion affects predictions of resilience and alternative stable states under different scenarios of human impacts. We found evenness across the functional groups of herbivores is increasingly important for reef recovery as fishing pressure increases, and recovery is promoted when algal communities begin with more turf than macroalgae. Our findings also highlight that models omitting functional groups that comprise real communities will miss crucial phenomena, such as loss of alternative stable states for browser-dominated communities, or continued presence of alternative stable states in grazer-dominated communities even when fishing pressure is removed entirely. This work advances our ability to understand and predict coral reef resilience and alternative stable states in the Anthropocene, and provides guidance for conservation policy on fishing practices.

## Author summary

We examine the question "how does functional group composition of herbivorous fishes and their food resources (algae) shape the resilience of coral reefs in

**Data availability statement:** There are no primary data in the paper. All materials are available at https://github.com/shaynasura/herbivore_coral_resilience_model and we archived our code on Zenodo (DOI: 10.5281/zenodo.15225119).

**Funding:** The author(s) received no specific funding for this work.

**Competing interests:** The authors have declared that no competing interests exist.

response to overfishing?" We did this using a mathematical model that explicitly incorporates functional groups of herbivorous fish (grazers, browsers) and their algal targets (turf, macroalgae). One key finding is that the community composition of herbivore functional groups modulates – and can qualitatively alter – the resilience of reefs when subject to human disturbance. Specifically, we found evenness of functions (grazing vs browsing) within herbivore communities is increasingly important for reef recovery as fishing pressure rises. Another novel finding is that reefs with grazer-dominated communities may persist in degraded algal-dominated states even if fishing pressure is entirely removed. Our findings highlight the importance of monitoring for skewed function in herbivore communities and implementing fishing policy to protect functional diversity and preserve coral reef resilience. Broadly, we show ecosystem response to anthropogenic stressors can be drastically impacted by community composition, and that functional ecology provides a pathway to understanding these impacts. Here, we find ecosystem resilience is determined by the relative abundance of functional groups of consumers and the resources they target, highlighting the importance of considering these groups in conservation and fisheries policy.

## Introduction

As anthropogenic disturbances continue affecting virtually every ecosystem globally [1,2], it is essential for ecologists to decipher the determinants of community resilience, which is the ability to both resist disturbance or recover following disturbance [3,4]. Coral reefs, which are among the most diverse ecosystems on earth, are degraded by coupled natural and anthropogenic disturbances [5,6], motivating research into attributes that govern their resilience. Empirical evidence strongly demonstrates that functional diversity of herbivorous fishes is essential to maintaining reef resilience [7–9] because they consume algae that compete with coral [10–12]. However, overfishing is a pervasive and global disturbance, leading to reduced biomass and functional diversity of herbivorous fishes [13–15]. As overfishing continues to impact functional diversity of herbivorous fishes in the Anthropocene, it is critical to understand effects on coral reef resilience.

Herbivorous fishes are often classified as grazers or browsers, among other functional groups [16], based on whether they consume algal turf versus macroalgae, respectively (*sensu* [17,18]). Healthy reefs typically have abundant coral and sparse, closely-cropped algal turf [19], which are algal communities <2 cm tall containing filamentous algae and juvenile or cropped macroalgae [20]. In contrast, degraded reefs have sparse coral and abundant algae, often macroalgae [5,8,9], which are erect algae >2 cm tall [17,18]. Coral can recover after disturbance when reefs are covered in turf algae because coral larvae can recruit within most algal turf assemblages [21]. In contrast, macroalgae can prevent coral recovery [22] both by inhibiting coral recruitment [12,23] and out-competing juvenile and adult coral via allelopathy, overgrowth, shading, and abrasion [23,24]. High functional diversity of herbivorous

fishes is key to reef resilience as grazers promote resistance to phase shifts (i.e., rapid shifts in community composition) by cropping turf, while browsers facilitate recovery following a shift by removing macroalgae [7,8,16,25,26]. Despite strong empirical support for this relationship, functional groups of fishes have only recently been incorporated into models of reef resilience [27].

A fundamental question regarding coral reef resilience is whether shifts to degraded states are smooth and reversible threshold responses versus sharp bifurcations indicating alternative stable states [28]. Two hallmarks of alternative stable states are bistability, when a community can exist in two different states for a single external condition (e.g., [29,30]), and hysteresis, when there are different thresholds (or tipping points) for transitions between states depending on the direction of change in an external driver (e.g., [28,30]). Given these two characteristics of alternative stable states (bistability, hysteresis), managing ecosystems that exhibit them is challenging since the threshold level for an external driver (e.g., fishing pressure) changes once a system shifts to a different state following a disturbance event [28]. Mathematical models are ideal tools to tackle this question, because they allow exploration of various initial reef states (e.g., high versus low coral cover) under similar environmental conditions (e.g., same reef system) in response to disturbances (e.g., fishing, cyclones) over long timescales, while these conditions are difficult to encounter naturally or manipulate empirically (but see [31] for an empirical example for two algal states).

Previous models provided novel insights into the myriad ways herbivory governs resilience and/or alternative stable states [27,30,32–38]. However, only recently have modelers begun to explore the impacts of functional diversity among herbivore or algal groups on reef responses to stressors [39], resilience [35], or alternative stable states [27]. Specifically, researchers [39] incorporated fish functional groups to examine short-term reef responses to anthropogenic stressors, but the functional groups were coarsely defined such that browsers and grazers were still combined as herbivores. Other researchers [35] explored how differential loss of algal functional groups influences reef recovery, but this was not explicitly linked to herbivore groups. Finally, researchers [27] included two herbivore functional groups (grazers and a key genus of browser) and linked them to algal functional groups (turf algae, immature macroalgae, mature macroalgae) to explore how selective fishing of these herbivorous reef fishes might impact reef resilience and alternative stable states. Together, these models revealed how consideration of even basic functional diversity can build understanding of reef ecosystem responses to stressors, with potential to provide science-based guidance to fisheries and conservation policy. This potential illustrates a critical need for further exploration of the roles of functional groups of herbivorous fishes, particularly grazers versus browsers versus generalists, in consumption of functional groups of algae and their importance in supporting coral reef resilience and influencing the occurrence of alternative stable states.

Here, we evaluate how functional groups of herbivorous fishes shape coral reef resilience and alternative stable states, building on growing themes in the empirical literature to develop new theoretical insights into reef ecology in the Anthropocene. We present a new model for coral reef communities that includes three herbivorous fish functional groups and their algal targets, and we use this framework to study how specific functional roles shape predictions of resilience and alternative stable states under different scenarios of human impacts (e.g., fishing). We examine fishing impacts in terms of both general overfishing and distinct herbivore communities arising from targeted protection, by varying overall fishing pressure and community dominance of herbivore functional groups, thus illuminating the crucial role of herbivore functional diversity in governing coral reef resilience to fishing disturbance. We ask three questions using our model framework: 1) How does the functional composition of the herbivorous fish community and total herbivory pressure (set by fishing pressure) affect reef recovery from disturbance? 2) How does dominance by different herbivore groups under different fishing pressures interact with abundance of their targeted resources to affect reef recovery after a disturbance? 3) How does variation in the dominant herbivore functional group influence the occurrence of alternative stable states in response to fishing pressure? Overall, we introduce a parsimonious model that captures essential inter-trophic relationships of this notoriously complex and diverse ecosystem, and gain valuable insights into how ecological community context and human impacts affect reef resilience.

# Results

We expanded a previous coral reef model from [36] (Fig A in S1 Appendix, Table A in S1 Appendix) to include herbivore functional groups (grazers, browsers, generalists) and the algal resources they consume (turf, macroalgae) (Fig 1, Table B in S1 Appendix). We model grazers and browsers as strict specialists on turf and macroalgae, respectively, while we include a generalist group to represent fishes that cross over between the grazing and browsing functions and consume turf and macroalgae in proportion to their abundance. We examined coral reef resilience (recovery after disturbance) and two hallmark characteristics of alternative stable states (bistability and hysteresis) to examine how herbivore functional groups influence coral reef community shifts from coral dominance to macroalgal dominance (Fig 2A). We analyzed the impacts of overfishing by varying overall fishing pressure among simulations and included the impacts of long-term protection of specific functional groups of herbivores by considering scenarios with different relative abundances of fish functional groups. This allowed us to assess how these human impacts affect coral reef resilience via the influence of different fish functional groups on algal dominance.

We found the functional composition of the herbivore community can play a dramatic role in mediating the risk of fishing pressure driving shifts in benthic communities from coral to macroalgae (Fig 2B, 2C). Even at relatively high fishing pressure, an herbivore community dominated by generalists (Fig 2C) or evenly split between herbivore functional groups (Fig Di in S1 Appendix) maintains a healthy, coral-dominated reef, while a community dominated by grazers allows a complete phase shift from coral to macroalgae (Fig 2C). An herbivore community dominated by browsers results in an intermediate scenario where coral cover is reduced, but still dominant, while cover of both turf and macroalgae persist at low levels (Fig Dii in S1 Appendix). For other fishing pressures and community compositions, see Fig D in S1 Appendix.

### Question 1: How does the functional composition of the herbivorous fish community and total herbivory pressure (set by fishing pressure) affect reef recovery from disturbance?

We simulated the aftermath of a disturbance by setting initial conditions to 15% coral cover and 70% turf algae, and found coral reef recovery is impacted dramatically by initial herbivore community composition (Fig 3, see Fig E in S1 Appendix for all scenarios). Overall, increased fishing pressure leads to more scenarios where coral does not recover (Fig 3 compare top to bottom). However, regardless of fishing pressure, recovery to high coral cover is more likely with increasing initial abundance of generalists (Fig 3 compare left to right), since they consume both turf and macroalgae, thus crossing over between grazing and browsing functions.

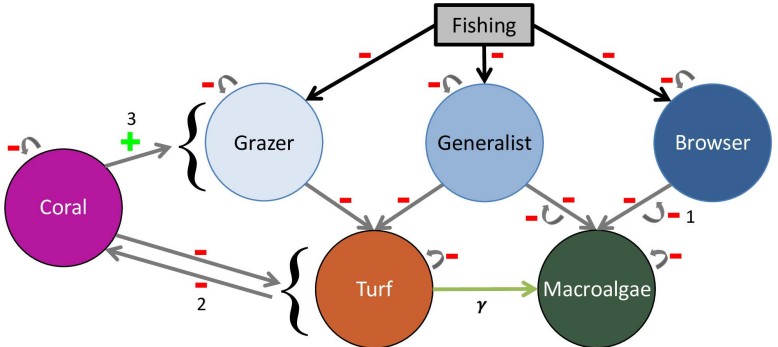

**Fig 1. Model Structure.** Diagram of our model structure that incorporates three herbivore functional groups (grazer, generalist, browser) and two algal resources (turf, macroalgae). Green arrow (γ) between "Turf Algae" and "Macroalgae" is a transition from turf to macroalgae. Our model incorporates the interactions and feedbacks (labeled 1–3 corresponding to their descriptions in S1 Appendix) from the [36] model.

**Fig 2. Shifts from Coral Dominance to Macroalgal Dominance.** A) Photos of a reef with high coral cover (left) versus with high macroalgal cover (right). The shift from high coral to macroalgal cover can result from increased fishing pressure and changes in the dominant herbivores present, which we explored with our model. Photos taken by Shayna Sura. B) Output of our model under 2 levels of fishing pressure (0.1, 0.9) for an even herbivore community and initial conditions and parameter values given in Table B in S1 Appendix. With increased fishing pressure, coral cover declines and macroalgal cover increases and persists over time. C) Output of our model under 2 herbivore community composition scenarios (generalist-dominated, grazer-dominated), for fishing pressure = 0.5. For a given fishing pressure, the benthic state of the reef can vary greatly depending on the herbivore community composition. Coral cover remains high for an herbivore community dominated by generalists, but macroalgal cover increases and persists over time for an herbivore community dominated by grazers, while a browser-dominated community leads to moderately high coral cover (Fig Dii in S1 Appendix). For both panels B and C, the Herbivores and Algal state variables are sums of their corresponding state variables (i.e., Herbivores = Grazers + Browsers + Generalists, and Alage = Turf + Macroalgae).

Reef recovery also occurs when the initial abundances of grazers versus browsers are more similar (band of brighter color in middle of triangles in Figs 3, and Fig Fi in S1 Appendix), again because both turf and macroalgae are being consumed. In contrast, a skew in the herbivorous fish community towards dominance by either grazers or browsers results in lower coral cover (light edges of triangles in Fig 3 panels), because one type of algae is under less top-down control. Although differences are more subtle, the reef is more likely to recover to higher coral cover if the skew favors browsers over grazers (compare light edges of triangles within Fig 3), which is due to browsers limiting macroalgae's stronger negative effect on coral compared to turf. When more grazers are present, they control the turf, but not the macroalgae.

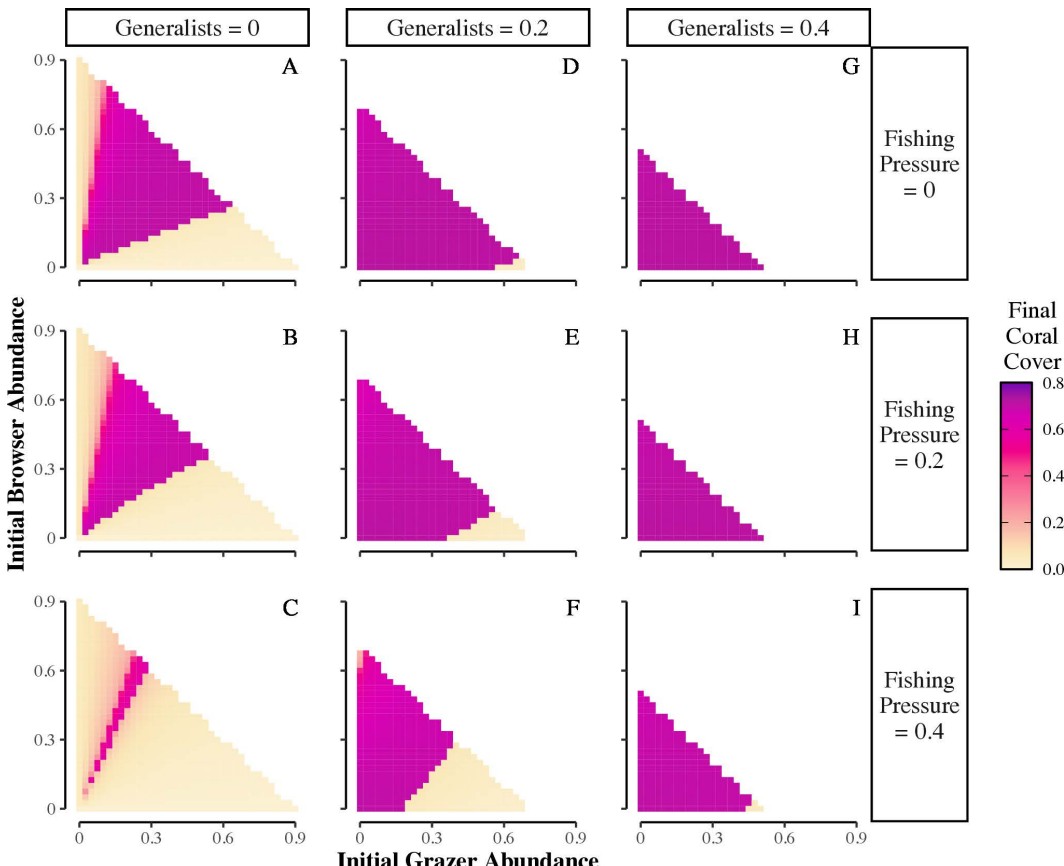

**Fig 3. Impacts of herbivore community composition and fishing pressure on coral reef recovery from disturbance.** Heatmaps showing how the initial abundance of herbivorous fishes that are grazers versus browsers affect final coral cover for fishing pressures ranging from 0 to 0.4 (rows) and for initial generalist abundance set to 0 (A-C), 0.2 (D-F), or 0.4 (G-I). Total herbivore abundance is restricted to ≤0.9; therefore, as initial generalist abundance increases, the maximum abundance of browsers+grazers has to decrease accordingly, resulting in smaller response spaces in the panels. Other initial conditions include: $C_{t=0}=0.15$, $T_{t=0}=0.7$, and $M_{t=0}=0$. Other parameter values: $i_C=0.05$, $i_T=0.05$, $i_M=0$, $b_C=0.3$, $b_T=0.8$, $b_M=0.5$, $d_C=0.1$, $g_T=2$, $g_M=1$, $r=1$, $\eta_T=0$, $\eta_M=1$, $a_T=0.25$, $a_M=0.5$, $\sigma=0.6$, and $\gamma=0.1$. Note: final coral cover values show very little variation for scenarios resulting in coral-dominated states, with 91% of final coral cover values >0.5 (coral dominated) within the range of 0.65 – 0.72, hence the similarity in colors.

We also studied the changes in algal community composition that underpinned these results. The reef shifted to macroalgae when generalists and browsers are sparse and grazers abundant (Fig 4, dark green triangles; Fig Fii in S1 Appendix). Without enough browsers or generalists, macroalgae expands and becomes dominant as grazers do not consume macroalgae. In contrast, the reef only remains turf-dominated in the rare circumstance when the herbivore community is dominated by strict browsers (Fig 4A-C, dark orange slivers along y-axes; Fig Fiii in S1 Appendix), as browsers only consume macroalgae and sparse generalists and grazers do not limit turf expansion. Increased fishing pressure increases the scenarios where reefs shift to macroalgae (Fig 4 compare top to bottom), as fishing pressure reduces browser and generalist abundance below a threshold where they can control macroalgal expansion.

Overall, we found reef recovery to high coral cover is promoted by communities skewed towards herbivores that consume macroalgae (browsers, generalists) over grazers. Abundant generalists promote coral recovery because both algal resources experience top-down control. However, a skew towards either grazers or browsers allows their non-targeted algal resource to expand faster than herbivory removes it, with increased fishing pressure exacerbating the outcome. Since macroalgae has a stronger competitive effect on coral compared to turf algae in our model, a skew towards grazers

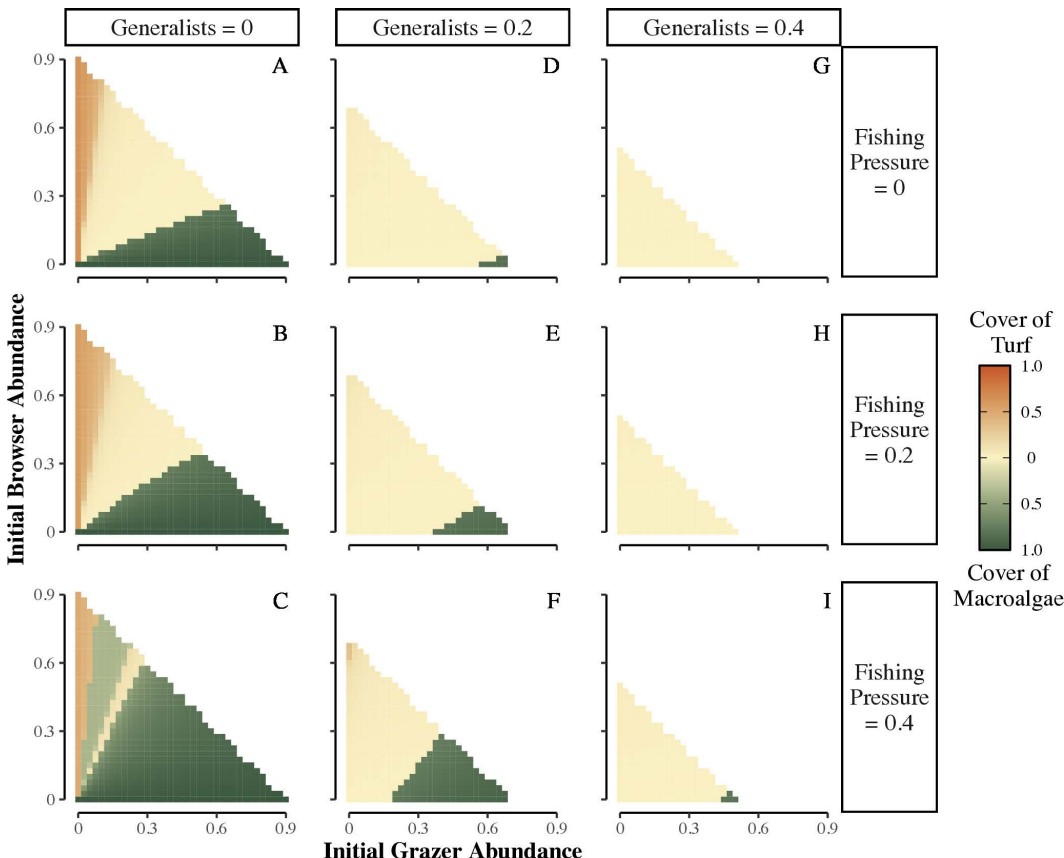

**Fig 4. Impacts of herbivore community composition and fishing pressure on algal cover after disturbance.** Heatmaps showing final cover of turf or macroalgae for the same scenarios as Fig 3. Specifically, these panels examine the initial abundance of grazers and browsers for fishing pressures ranging from 0 to 0.4 (rows) and for initial generalist abundance set to 0 (A-C), 0.2 (D-F), or 0.4 (G-I). Other initial conditions include: $C_{t=0} = 0.15$, $T_{t=0} = 0.7$, and $M_{t=0} = 0$. See Fig 3 caption or Table B in S1 Appendix for other parameter values. For each scenario, the heatmap indicates the final cover of turf or macroalgae, depending on which had the higher final cover. Note that these results are complementary to the level of coral cover shown in Fig 3; specifically, turf and macroalgal cover values near zero (beige color) correspond to high coral cover values shown in Fig 3.

and more macroalgal expansion will have a more detrimental effect on coral recovery than a skew towards browsers and more turf expansion. Generally, browsers are considered more important than grazers for reef recovery from shifts to macroalgal dominance [16,26]. However, our results also indicate browsers can be more important than grazers for prevention of macroalgal dominance, and both browsers and generalists better promote recovery of reefs with high turf algal cover after a disturbance compared to grazers.

**Question 2: How does dominance by different herbivore groups under different fishing pressures interact with abundance of their targeted resources to affect reef recovery after a disturbance?**

The ability of the herbivore community to facilitate coral recovery is maximized by low fishing pressure, the presence of generalists, and a sparse macroalgal community (Fig 5). Overall, recovery decreased as herbivorous fish communities changed in dominance from generalists (Fig 5G-I) to browsers (Fig 5A-C) to grazers (Fig 5D-F), and recovery was negatively impacted by increased fishing pressure (Fig 5 compare rows from top to bottom, see Fig G in S1 Appendix for all scenarios). While all scenarios started with an initial disturbance reducing coral cover to 15%, the dependence of coral recovery on initial benthic algal cover varied strongly depending on herbivore community composition. In a

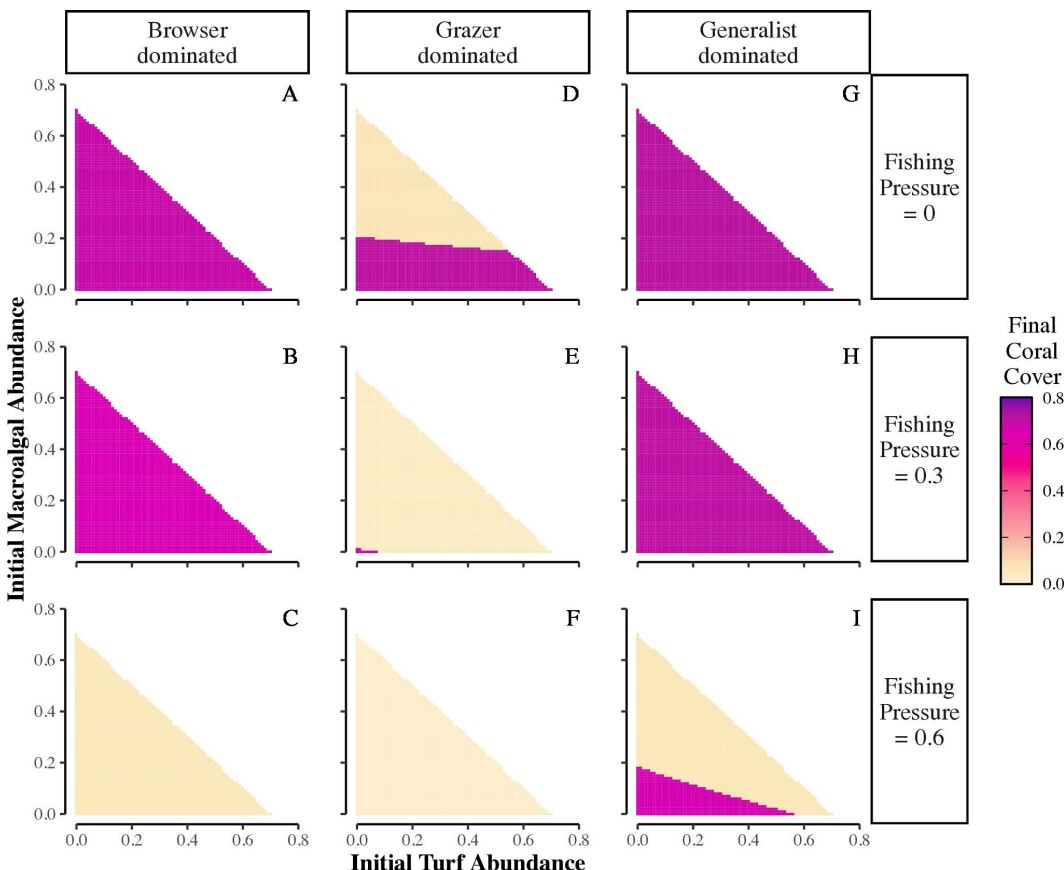

**Fig 5. Interaction of turf and macroalgal dominance and herbivore community composition on coral reef recovery from disturbance.** Heat-maps showing how initial cover of turf and macroalgae affect final coral cover for fishing pressures ranging from 0 to 0.6 (rows) and for initial herbivore proportions set to 1 of 3 scenarios: 1) **browser-dominated** (A-C): browsers=0.6, generalists=0.15, grazers=0.15; 2) **grazer-dominated** (D-F): grazers=0.6, generalists=0.15, browsers=0.15, or 3) **generalist-dominated** (G-I): generalists=0.6, grazers=0.15, browsers=0.15;. Total initial benthic cover is restricted to ≤0.85, and initial coral cover ($C_{t=0}$) = 0.15; therefore, the amount of unoccupied space changes for each combination of initial turf and macroalgae abundance. See Fig 3 caption or Table B in S1 Appendix for other parameter values.

generalist-dominated community, the reef recovers to high final coral cover for low and intermediate fishing pressures, regardless of initial amounts of turf or macroalgae (Fig 5G,H). However, for high fishing pressure (0.6), this community can only recover in an unlikely scenario, when initial turf and macroalgae are both low so there is ample open space for coral recovery (Fig 5I). Also, in this scenario, coral reef recovery occurs for a wider range of initial turf cover than macroalgal cover.

In contrast, reef recovery in a browser-dominated community shows no dependence on initial cover of turf versus macroalgae, and fishing pressure is the most important factor (Fig 5A-C). Finally, when grazers dominate the herbivore community, reef recovery depends sharply on the initial cover of macroalgae when fishing pressures are low (≤0.2, Figs 5D, and G in S1 Appendix). Since our modeled grazers only consume turf, if the reef starts with too much macroalgae, it can escape top-down control if the community of generalists and browsers is too sparse, which prevents reef recovery. The threshold of macroalgal cover that prevents coral recovery decreases as fishing pressure increases from 0 to 0.2, while initial turf cover has relatively little effect on reef recovery (Fig G in S1 Appendix).

## Question 3: How does variation in the dominant herbivore functional group influence the occurrence of alternative stable states in response to fishing pressure?

Herbivore functional group dominance governs the existence and nature of alternative stable states (Fig 6A-D). Bistability is evident when, for a given fishing pressure, both low (<0.2) and high (>0.6) final coral covers occur, depending only upon the initial coral cover. When the herbivore community is dominated by generalists, there is a moderate amount of bistability in relation to fishing pressure (Fig 6B), similar to when the community has equal abundances of herbivore functional

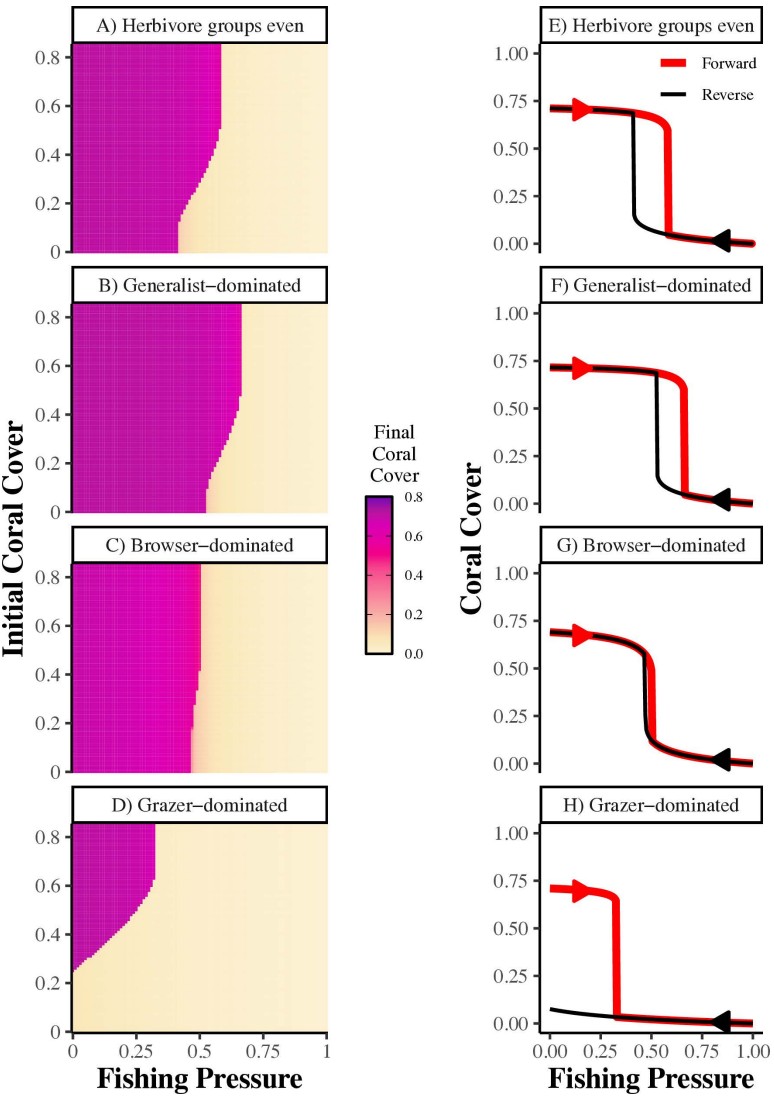

**Fig 6. Impacts of herbivore community composition on coral reef bistability and hysteresis.** Bistability plots (A-D) showing final coral cover for each combination of fishing pressure and initial coral cover conditions. Bistability is evident when, for a given fishing pressure, both low (<0.2) and high (>0.6) final coral covers occur, depending only upon the initial coral cover. Hysteresis plots (E-H) showing the final coral cover as fishing pressure is increased (forward) or decreased (reverse). Hysteresis is evident when there is a range of fishing pressures where the forward (red) and reverse (black) lines do not overlap. We examined four scenarios using our expanded model parameter values from Table B in S1 Appendix with different initial herbivore community compositions: **herbivore groups even** (A,E): generalists = grazers = browsers = 0.3; **generalist-dominated** (B,F): generalists = 0.6, grazers = 0.15, browsers = 0.15; **browser-dominated** (C,G): browsers = 0.6, generalists = 0.15, grazers = 0.15; or **grazer-dominated** (D,H): grazers = 0.6, generalists = 0.15, browsers = 0.15. See Fig 3 caption or Table B in S1 Appendix for other parameter values.

groups (Fig 6A). Also, the generalist-dominated scenario can sustain the highest level of fishing pressure and still maintain high coral cover (Fig 6B). In contrast, the browser-dominated scenario has only a sliver of bistability, resembling more of a threshold transition (Fig 6C). This finding arose when browsers initially constituted 60% of the herbivore community; we explored higher initial browser abundances and found total loss of bistability and a threshold transition for initial browser abundance of 70% or higher (Fig J in S1 Appendix). Also, when browsers dominate, final coral cover is slightly lower across all fishing pressures compared to when generalists dominate (Fig 6B,C – compare darkness of magenta). Meanwhile reefs dominated by grazers have the widest region of bistability, and, thus, are susceptible to disturbance-induced shifts to low coral cover for the broadest range, and lowest values, of fishing pressures (Fig 6D). There is bistability even with no fishing pressure, where recovery to high coral cover can only occur if disturbance does not reduce coral cover to <0.27 (for the illustrative parameters chosen here). Thus, building on findings from Fig 5, our model shows that reefs dominated by grazers can only recover from moderate disturbances that result in turf dominance.

As expected from theory, the presence of hysteresis follows the same patterns as bistability, so the existence and range of hysteresis is driven by herbivore community composition (Fig 6E–H). Generalist-dominated reefs (Fig 6F) maintain high coral cover across a wider range of fishing pressure and exhibit hysteresis at pressures similar to reefs with even herbivore communities (Fig 6E). When browsers dominate, hysteresis is narrow and the transition approaches a threshold response (see Figs 6G and K in S1 Appendix for threshold scenarios arising at higher levels of browser dominance). Lastly, grazer-dominated communities exhibit hysteresis for the widest range of fishing pressures, and once the system is degraded, it cannot recover to high coral cover even if fishing pressure is eliminated completely (Fig 6H).

## Discussion

Our findings highlight the crucial importance of maintaining a balance of coral reef fishes' functional groups to foster coral reef recovery, and that lack of protection of critical groups will leave coral reefs vulnerable to shifting to degraded community states from which they cannot easily recover (i.e., alternative stable states). We accomplished this by leveraging recent empirical advances to explicitly incorporate herbivorous fish functional groups and their algal targets into our model framework. One key finding from our model is that the relative abundance of herbivore functional groups can modulate – and sometimes qualitatively alter – the resilience of reefs as well as the existence and strength of alternative stable states in response to human disturbance. By using our model to explore the impacts of generalized overfishing (via increased fish mortality, which reduces overall herbivory pressure) and the consequences of long-term fishing or conservation practices (via scenarios with different herbivore community composition), we provide insight into how reef recovery is influenced by broad anthropogenic pressure and the loss of particular herbivore functional groups, which could not be easily revealed empirically.

### Functional composition of herbivore communities can modulate the resilience and alternative stable states of coral reefs

We found the resilience of coral reefs and occurrence of alternative stable states for reef ecosystems can be impacted dramatically by the balance of functional groups in the herbivore community. Reefs with herbivore communities dominated by grazers are the least resilient to fishing pressure and were unable to recover to coral dominance even after fishing ceased. Browser-dominated communities are more resilient than grazer-dominated communities and have the added benefit of lacking alternative stable states in some cases. Finally, we found generalist-dominated communities (or communities with balanced functional composition) are the most resilient since all herbivory functions are performed. Our findings align with numerous recent empirical studies of reef systems. An analysis of 601 reef sites spanning a decade found that co-occurrence of herbivore functional groups was correlated with reduced algal cover and increased coral cover [40]. Other field studies have shown that reef recovery from disturbance is promoted by high diversity and evenness of herbivorous fish communities on coral reefs (diversity: [8,9,25]; evenness: [41]), and by predator diversity in kelp forests [42].

Developing a model framework that represents different functional groups of herbivorous fishes enabled us to expand on these empirical studies by exploring how specific herbivore roles and communities shape reef resilience or the role of alternative stable states.

Human impacts in the form of generalized and targeted overharvesting of herbivorous fish increase the importance of understanding how herbivore functional diversity supports coral reef resilience. Data show that fishing pressure can differentially affect herbivore functional groups [15,27], which can skew herbivore community compositions and negatively impact current benthic reef state [40] and prevent reef recovery (e.g., [22]). Our model framework allowed us to expand on empirical findings by examining how both generalized overfishing (modeled as additional mortality for all fish groups) and targeted fishing practices (modeled as scenarios with different relative abundances of functional groups) influence reef resilience and the occurrence of alternative stable states. We found the herbivorous fish community that best promotes reef recovery from disturbance changed with fishing pressure intensity, with evenness and preservation of the browsing function becoming increasingly important as generalized fishing pressure increased. This broad finding was also reported by [27] using a different model structure, highlighting its robustness and underscoring the timely nature of this work. As human impacts, specifically overfishing, continue in the Anthropocene, it is especially important to monitor for skewed function in herbivore communities and implement or restrict targeted fishing on certain groups in coral reefs to preserve reef resilience. This finding could not have been uncovered using empirical data or prior models lacking explicit herbivore functional groups.

## Grazer-dominated communities may erode coral reef resilience

We found reefs with herbivore communities skewed towards grazers became dominated by macroalgae for lower levels of fishing pressure compared to other herbivore communities; that is, they have lower resilience to overfishing. Empirical work has demonstrated that overall low abundances of grazers and browsers are strongly associated with phase shifts to macroalgal-dominance [8], but the relative contribution of these two groups could not be distinguished. Both our study and an independent model incorporating herbivore functional diversity [27] identified that overfishing of browsers can play a major role in loss of coral reef resilience.

However, whereas [27] found that reefs with grazer-dominated communities (represented in their model by highly selective harvesting of unicornfish, their exemplar of browsers) have the smallest region of bistability, we found reefs with grazer-dominated communities have the widest region of bistability in response to fishing pressure. Further, in our model, dominance by grazers may prohibit recovery from large disturbances that reduce coral cover below a certain threshold, even if fishing is completely absent. One likely explanation for this contrast is that our model assumes herbivore abundance is primarily regulated by shared shelter resources (coral cover), while [27] model herbivore abundance with separate carrying capacities for each group and population growth rates governed by algal resources (including macroalgal cover for browsers). Thus, in [27], when the community is dominated by grazers, lack of consumption of macroalgae allows this food resource to become more abundant, enabling browser populations to increase and exert stronger top-down control on the macroalgae, stimulating coral recovery. In contrast, in our model, high abundance of grazers does not result in an ultimate rebound of the browser population because the limiting resource of shelter is shared, so browsers remain scarce and the reef remains dominated by macroalgae. Taken together, we hope these studies motivate more research on the relative importance of food and shelter resources to the recovery of browser populations, as the contrast between these models identifies this as a critical need to understand reef resilience.

Empirical studies comparing effects of grazer- versus browser-dominated communities on reef resilience are rare because it would be logistically difficult and unethical to manipulate herbivore communities on a large scale. However, a recent empirical study observed a set of recently disturbed patch reefs where grazers dominated the community, and quantified the degree of coral recovery versus persistence of macroalgal domination while manipulating initial macroalgal cover or herbivory pressure across patch reefs [22]. Our model findings align with their empirical findings that the

grazer-dominated community could not promote coral recovery when overall herbivory pressure was reduced (akin to increased fishing pressure in our model) or when the patch reefs started with high macroalgal cover [22]. The findings from [27] suggest coral recovery from macroalgal dominance is possible with an herbivore community dominated by grazers; however, this did not occur in the empirical study. One possible explanation for this contradiction is that the ambient herbivore community in the empirical study may experience a higher amount of fishing pressure than the low level needed to allow coral recovery in the [27] model. Overall, the findings from these three studies indicate further research is necessary to understand the role of grazers on coral reef resilience and alternative stable states.

### Browsers play a disproportionately important role, both qualitatively and quantitatively

Our analysis reveals a new hypothesis that coral reefs with herbivore communities dominated by strict browsers may not be susceptible to alternative stable states. Along with being more resilient to fishing pressure compared to grazer-dominated communities, we found reefs with strongly browser-dominated communities could exhibit simple threshold transitions as opposed to bistable regions. Thus, the browser-dominated scenarios are unique and contrast with reefs dominated by generalist or grazer herbivores, which do exhibit bistability. Alternative stable states can be problematic for managing ecosystems because a greater reduction in the external driver (e.g., fishing pressure) compared to the original tipping point is necessary to recover from phase shifts in response to disturbance events [28]. Thus, lack of alternative stable states for browser-dominated communities could make it easier for fisheries management to identify critical thresholds and more easily reverse phase shifts when they occur.

Because we modeled our browsers as strict specialists on macroalgae while [27] modeled their browsers akin to our generalists (i.e., consuming both turf and macroalgae), the lack of alternative stable states in this scenario is a new finding with no prior points of comparison. We believe it is driven by our assumption that herbivore populations are regulated by competition for a shared resource, and particularly by the limiting resource of coral cover to provide shelter. When browsers are a small proportion of the total herbivore community, they are limited by the abundance of grazers and generalists via the overall herbivore carrying capacity, which in turn is linked primarily to coral cover. The resulting limited effect of fishing pressure on browser abundance can lead to hysteresis, and hence bistability under generalist-dominated and grazer-dominated scenarios. To illustrate this, consider a grazer-dominated reef that experiences elevated fishing pressure and degrades to a macroalgal-dominated state with low coral cover – the low coral cover leads to low total herbivore carrying capacity, which is largely occupied by grazers. In this circumstance, browsers cannot increase despite the abundant supply of macroalgae, so they cannot exert top-down control on macroalgae. This remains true even if fishing pressure is removed, since coral cover remains low, leading to hysteresis. In contrast, for reefs dominated by browsers, any fishing pressure, even generalized across all herbivores, exerts direct control on browsers, which in turn exert direct control on macroalgae. As fishing pressure increases, browsers decrease, and macroalgae increase; if fishing pressure decreases, the reverse occurs with no hysteresis. This allows a simple threshold transition between coral and macroalgal dominance of the reef related to fishing pressure. Overall, the qualitative loss of alternative stable states for some herbivore community scenarios could be a critical finding for the conservation and management of fisheries in coral reef ecosystems and warrants further investigation into the underlying assumptions and dynamics.

Although the lack of alternative stable states may ease fisheries management concerns, our model exhibits a tradeoff as having a browser-dominated community also pushes the threshold transition between high and low coral cover states to a lower level of fishing pressure compared to an even herbivore community. This finding appears to contrast with [27] who observed tipping points at higher levels of fishing pressure for communities with the browsing function increasingly represented; however, the 'browsers' in their model are more similar to the generalists in ours, which also exhibit higher tipping points. For our model, the lower tipping point for strict browsers can manifest due to limited control of turf algae when browsers are dominant. Thus, a lower amount of fishing pressure is sufficient to reduce the generalist and grazer populations such that they lose control of turf algae. Although turf algal cover eventually transitions to macroalgae that can

be consumed by browsers, while in the turf state they can still outcompete coral for benthic space and cause a transition to low coral cover once fishing pressure reduces herbivory pressure sufficiently.

Our finding that browsers play a more critical role than grazers, even on turf-dominated reefs, raises new perspectives on how to interpret empirical data. Browsers have long been recognized as more important than grazers for recovery after a reef has shifted to macroalgal dominance [16,26]. However, even in scenarios initiated with high turf cover, we found browsers remained more important than grazers for recovery from turf dominance. This appears inconsistent with empirical data showing that, in the presence of an herbivorous fish community containing few browsers [43–45], coral recovered from turf dominance following recent disturbances, suggesting a community containing more grazers than browsers can enable reef recovery from turf dominance. This apparent disconnect between model and empirical data may occur because we modeled our grazers and browsers as complete specialists, hence grazers in our model do not consume any macroalgae. However, some herbivores classified as grazers are observed to consume macroalgae [46–48] in addition to turf algae, suggesting they may be better represented by our modeled generalists. For our model scenarios with even a small proportion of generalist herbivores, having the rest of the herbivore community dominated by strict grazers can still result in reef recovery (Fig 3D), which aligns with empirical evidence. Thus, our findings suggest that highly specialized herbivore communities may reduce coral reef resilience, and better characterizing the degree of specialization versus generalization of herbivorous reef fish is a key research frontier.

## Generalists buffer the system and support reef resilience

A balance in herbivory functions is crucial for reef resilience, whether it comes from having a balanced community of specialists or from having enough generalist herbivores. Our finding that the most resilient reefs are those with herbivore communities performing all herbivory functions (e.g., generalist-dominated) aligns with empirical findings [8,9,25,41], and with modeling work by [27]. Our finding also aligns with empirical data indicating a reef has more coral when herbivorous fish functional groups co-occur in similar abundances [40]. We modeled browsers and grazers as complete specialists, and generalists as fishes that consumed turf and macroalgae in proportion to their availability. Thus, our model framework enabled a distinct examination of the grazing versus browsing functions of herbivores, while also directly examining the importance of generalists (e.g., a crossover group consuming both turf and macroalgae). Herbivorous fishes classified as browsers and grazers are commonly observed foraging beyond their functional group classification [46–48]; hence, they are not considered complete specialists. However, the context-dependency of herbivorous foraging behavior based upon what food is available is rarely studied (but see [49]), which hinders our understanding of where herbivorous fishes align along a continuum of specialists versus generalists. Our model suggests additional research into the foraging behavior of herbivorous fish, and a framework for classifying them along a specialist-generalist spectrum may be valuable for assessing coral reef herbivorous fish communities and their contributions to reef resilience.

## Model Assumptions Motivating Future Work

Coral reefs are some of the most diverse ecosystems on earth, so assumptions are required to model them parsimoniously. We previously discussed the value of investigating the implications of how herbivore populations are tied to shelter versus algal resources, and how to model herbivore groups based on their specialization and selection of resources. Our model also assumes that herbivore functional groups have the same population growth rates (which are not coupled to food resources), experience the same mortality rates from fishing pressure, and are equally dependent upon coral cover for their carrying capacity. In reality, different fish species have variable life histories, are differentially targeted by fishers, and vary in their dependence on coral for shelter and habitat. Our parsimonious formulation means that herbivore dynamics in our model are very simple, and relative abundances of the functional groups are determined by initial conditions. This approach is sufficient for the questions and scenarios we address here, but future work could explore the impacts of these ecological differences – which would be especially important if studying selective fishing practices – by modifying

the elements of our model that represent herbivore population ecology (e.g., by using different *r, f,* and σ values for each herbivore group, by coupling population growth rates to trophic conditions, or by introducing a degree of group-specific population regulation). Notably, however, [27] made different assumptions and addressed some of the above heterogeneities and reached parallel conclusions on many key points, providing at least partial support for the robustness of our core findings to these assumptions. Additionally, future models could be expanded to examine additional functional groups of herbivores (scrapers/small excavators, large excavators/bioeroders) or additional categories of coral (e.g., juvenile, adult), which could have distinct interactions with turf and macroalgae. Finally, as coral reefs face increasing cumulative impacts from stressors (marine heat waves, ocean acidification, nutrient pollution) in the Anthropocene [2], future work should incorporate multiple stressors and examine their interactive effects on resilience.

## Conclusions

We add to a growing body of evidence that incorporating functional groups and trophic structure into models can yield important advances in understanding and predicting ecosystem dynamics for coral reefs (e.g., [50,51]) and other systems (e.g., [52–54]). Our work supports and extends recent empirical and modeling evidence of the importance of herbivore functional groups for coral reef resilience and occurrence of alternative stable states. Our findings highlight the importance of considering herbivore functions for management of these resources, specifically because skewed herbivore communities may result in complete loss of alternative stable states for browser-dominated reefs, or the continued presence of alternative stable states even when fishing pressure is removed entirely for grazer-dominated reefs. We further identify novel hypotheses for future theoretical work and research priorities for future empirical studies. Our work captures the essence of a famously diverse ecological community by adopting a functional perspective within a parsimonious model framework, which is a foundation for predicting responses and informing solutions to human impacts that will only accelerate in the Anthropocene.

## Methods

We expanded a previous coral reef model to incorporate three herbivore functional groups and the two algal groups upon which two of them specialize. We examined coral reef resilience (recovery after disturbance) and characteristics of alternative stable states (bistability and hysteresis) under varying levels of fishing pressure, compositions of herbivorous fish, and benthic communities.

*Model Structure* – Our model was based on [36] (Fig A and Table A in S1 Appendix), which we reconstructed with identical results for a range of fishing pressures (Figs B, C in S1 Appendix; see S1 Appendix for a full description of our recreation and basic analyses).

We expanded their model to include herbivore functional groups (grazers, browsers, generalists) and the algal resources they consume (turf, macroalgae) (Fig 1, Table B in S1 Appendix). We define turf as fast-growing filamentous algae and juvenile or cropped macroalgae [20] and macroalgae as erect algae >2 cm tall [17,18]. Grazers are a functional group that specialize on turf while browsers specialize on macroalgae [16]. To isolate the precise roles of these foraging behaviors, we assumed complete specialization of these two groups. However, because many herbivorous fishes may not be entirely specialized [46–48,55–57], we included a group for generalists, which represent fishes that can cross over categories by consuming both turf and macroalgae. For simplicity, we assumed these crossover fishes consume the two algal types in proportion to their availability.

Our model is continuous time and deterministic, has state variables with density-dependent negative feedbacks to prevent unlimited growth, and includes 3 de-stabilizing positive feedbacks: 1) herbivory rate increasing with algal cover via a Holling type II functional response, 2) negative effects of algal cover on coral recruitment and growth, and 3) herbivore carrying capacity increasing with coral cover via provision of shelter (see Table C in S1 Appendix for references from [36] supporting this feedback).

Each benthic state variable is modeled as the proportion of space occupied, with coral ($C$), turf ($T$), and macroalgae ($M$) competing for benthic space; the remainder is unoccupied space ($S$).

$$S = 1 - C - T - M \qquad (1)$$

Coral ($C$), turf ($T$), and macroalgae ($M$) expand into unoccupied space through import of propagules ($i_C$, $i_T$, $i_M$) and expansion of existing cover ($b_C$, $b_T$, $b_M$). Expansion of coral is negatively affected by the competitive effect of turf and macroalgae ($\alpha_T$, $\alpha_M$) on coral recruitment and growth, which is proportional to their cover. Coral decreases via a constant decay rate ($d_C$) representing coral mortality.

$$\frac{dC}{dt} = (i_C + b_C C)\, S(1 - (\alpha_T T + \alpha_M M)) - d_C C \qquad (2)$$

Turf can experience uncontrolled growth and transition into macroalgae at a rate of $\gamma$, when it is not cropped by grazer and/or generalist herbivores. Herbivores (grazers ($G$), browsers ($B$), generalists ($R$)) negatively affect turf and macroalgae via consumption, but there is a positive feedback between turf and/or macroalgae and herbivory rate, which is incorporated as a Holling type II functional response. Thus, herbivory rate on turf ($g_T$) and/or macroalgae ($g_M$) saturates with increasing turf and/or macroalgae and is affected by the handling time of turf ($\eta_T$) and/or macroalgae ($\eta_M$) for herbivores.

$$\frac{dT}{dt} = (i_T + b_T T)\, S - \gamma T - \left( \frac{g_T T G}{g_T \eta_T T + 1} + \frac{g_T T R}{g_T \eta_T T + g_M \eta_M M + 1} \right) \qquad (3)$$

$$\frac{dM}{dt} = (i_M + b_M M)\, S + \gamma T - \left( \frac{g_M M B}{g_M \eta_M M + 1} + \frac{g_m M R}{g_T \eta_T T + g_M \eta_M M + 1} \right) \qquad (4)$$

The relative abundance of each herbivore functional group ($G, B, R$) is modeled as a proportion of a shared herbivore carrying capacity, reflecting intra- and intergroup competition for limiting resources. Each functional group grows at rate ($r$) and decreases via fishing pressure ($f$), which is represented as a constant per capita loss rate. Coral positively affects herbivores by providing habitat and shelter (see [36] for justification); therefore, herbivore carrying capacity is affected by coral cover in proportion to the parameter $\sigma$, in addition to other limiting resources with weight ($1 - \sigma$). As $\sigma$ increases, coral increasingly influences herbivore carrying capacity. Authors of [36] provide numerous references supporting the positive feedback between corals and herbivores (Table C in S1 Appendix).

$$\frac{dG}{dt} = rG \left( 1 - \frac{(G + R + B)}{(1 - \sigma) + \sigma C} \right) - fG \qquad (5)$$

$$\frac{dB}{dt} = rB \left( 1 - \frac{(G + R + B)}{(1 - \sigma) + \sigma C} \right) - fB \qquad (6)$$

$$\frac{dR}{dt} = rR \left( 1 - \frac{(G + R + B)}{(1 - \sigma) + \sigma C} \right) - fR \qquad (7)$$

*Model Analyses* – We first confirmed that our output matched the [36] model, when we used initial conditions and parameter values that collapsed our model down to theirs (Table B – collapsed model in S1 Appendix). We ran both models for

1000 years (to reach equilibrium) under fishing pressures of 0.1, 0.5, and 0.9 per year, and compared changes in herbivores ($H$ in their model vs. $G+B+R$ in ours) and coral ($C$ in their model vs. $C$ in ours) and algal ($A$ in their model vs. $T+M$ in ours) cover over time.

For our main analyses, we addressed three questions with our model. For our first two questions, we explored a crucial aspect of coral reef resilience: the ability of coral to recover following a disturbance. For the third question we characterized the occurrence of alternative stable states in our system. We also explored the role of varying initial cover of coral, turf, and macroalgae following a disturbance, but found relatively little effect on reef recovery (see S1 Appendix "Supplemental Question 1").

We selected parameter values to reflect a coral reef system with multiple herbivore functional groups and their targeted algal resources, based upon the following assumptions:

- No handling time for turf algae ($\eta_T = 0$).

- Expansion of macroalgae is slower than expansion of turf ($b_M < b_T$).

- Macroalgae grow from recruits present in turf ($i_M = 0$).

- Mortality due to herbivory is greater for turf than macroalgae ($g_T > g_M$).

- Macroalgae exert a stronger competitive effect on coral than turf ($a_M > a_T$).

Based on these assumptions, and drawing on prior empirical and theoretical studies, we selected a base set of parameter values for our simulations (Table B "Model Values" in S1 Appendix). Evidence supporting these assumptions is detailed in the S1 Appendix. For some parameters, empirical evidence is not sufficient to support precise parameter estimates, or parameter values are expected to vary across ecological and geographic contexts. To explore the robustness of our findings to alternative choices for these parameter values, and to support application of our work to new settings where different parameters may apply, we conducted extensive sensitivity analyses showing how benthic cover outcomes (Figs M – R in S1 Appendix) and the occurrence of bistability (Figs S – BB in S1 Appendix) are shaped by the values of nine key parameters, across a range of fishing pressures, herbivore community compositions, and initial conditions. As expected, the system's steady state and bistability in a given simulation depends on the parameter values chosen, but the observed variation is consistent with the findings and ecological interpretations in the main study.

Unless stated otherwise, for all analyses, we used the model parameter values and initial conditions from Table B in S1 Appendix, ran the model for 1000 years to reach equilibrium (hereafter stated as steady state), and presented final coral cover for each analysis.

## Question 1: How does the functional composition of the herbivorous fish community and total herbivory pressure (set by fishing pressure) affect reef recovery from disturbance?

To simulate a major disturbance, we set initial cover to 0.15, 0.7, and 0, for coral, turf, and macroalgae, respectively, which left the remaining 0.15 as open space. We ran simulations where we varied initial abundance of grazers and browsers (0 – 0.9 in 0.025 increments) for initial generalist abundance set to 0, 0.2, or 0.4. Total initial herbivore abundance (grazers + browsers + generalists) was not held constant in these simulations; rather it ranged from 0 to 0.9 of carrying capacity so we could assess how differences in the total initial herbivore abundance (due to previous fishing pressure) affected final coral cover. Also, we set total initial herbivore abundance $\leq 0.9$, to ensure herbivore populations began below their carrying capacity. We set fishing pressure to 0 – 0.7 of herbivore abundance in increments of 0.1. We present scenarios with fishing pressure = 0, 0.2, and 0.4 in the main text figures, and present all scenarios in the S1 Appendix. For this question, we summarize results as final coral cover and final cover of the dominant algal group (turf or macroalgae).

**Question 2: How does dominance by different herbivore groups under different fishing pressures interact with abundance of their targeted resources to affect reef recovery after a disturbance?**

We varied initial turf and macroalgal cover for initial herbivore abundance set to three scenarios: 1) generalist-dominated, 2) browser-dominated, or 3) grazer-dominated. For these scenarios, the dominant group = 0.6 and other groups = 0.15. We set initial coral cover at 0.15 for all simulations and varied initial turf and macroalgal cover from 0 – 0.7. As above, we constrained the total initial benthic cover ≤0.85. We set fishing pressure to 0 – 0.7 in increments of 0.1. We present scenarios with fishing pressure = 0, 0.3, and 0.6 in the main text figures (see S1 Appendix for all scenarios).

**Question 3: How does variation in the dominant herbivore functional group influence the occurrence of alternative stable states in response to fishing pressure?**

Two hallmarks of alternative stable states are bistability and hysteresis. Bistability in our model is evidenced when final coral cover is split between high (>0.6) and low (<0.2) coral equilibrium states for the same level of fishing pressure, depending upon initial coral conditions. For example, consider a reef that has 70% equilibrium coral cover ($C = 0.7$) when exposed to moderate fishing pressure of 0.3. Then this reef experiences an extrinsic disturbance, such as a storm or warming event, that reduces coral cover to 20%. If the reef returns to a high coral cover equilibrium state while the same amount of fishing pressure is maintained, it does not exhibit bistability. However, the reef does exhibit bistability if it persists in a low coral cover equilibrium state following the disturbance while the same amount of fishing pressure is maintained.

We compared levels of fishing pressure resulting in bistability for different scenarios of herbivore community dominance and initial coral cover (modeling different levels of disturbance). To do this, we examined combinations of initial coral cover and fishing pressures ranging from 0 – 0.85 and 0 – 1, respectively, in increments of 0.01. After subtracting initial coral cover and empty space (0.15) from total benthic space available, remaining space was evenly split between turf and macroalgae for their initial values. For each combination of initial coral cover and fishing pressure, we ran the model to steady state and examined final coral cover. First, we examined bistability using parameter values from Table B in S1 Appendix and equal initial abundances of the herbivore functional groups (i.e., generalists = grazers = browsers = 0.3). Then, we examined bistability for the 3 scenarios of herbivore abundances as above: 1) generalist-dominated, 2) browser-dominated, and 3) grazer-dominated.

To examine the model for hysteresis, we did "forward" and "reverse" simulations, reflecting increasing or decreasing fishing pressure, respectively. Then we determined whether transitions from high to low coral cover states differed based on direction of change in this environmental driver. Specifically, for the "forward" and "reverse" simulations, we increased then decreased fishing pressure from 0 – 1 in 0.005 increments, respectively. For each simulation, we used the end conditions for the state variables from the previous simulation for the next simulation. We examined hysteresis for the same four scenarios as for bistability.

## Supporting information

**S1 Appendix. Additional methods, results, tables, and figures, including sensitivity analyses.**
(PDF)

## Acknowledgments

This work was originally conceived as a project in a modeling course (EEB 219B) taught by JOL-S. We are grateful to Ana Gomez for her advice and guidance to SAS as Teaching Assistant for this course.

## Author contributions

**Conceptualization:** Shayna A. Sura, James O. Lloyd-Smith, Peggy Fong.

**Data curation:** Shayna A. Sura.

**Formal analysis:** Shayna A. Sura.

**Investigation:** Shayna A. Sura.

**Methodology:** Shayna A. Sura, James O. Lloyd-Smith, Peggy Fong.

**Supervision:** James O. Lloyd-Smith, Peggy Fong.

**Visualization:** Shayna A. Sura.

**Writing – original draft:** Shayna A. Sura.

**Writing – review & editing:** Shayna A. Sura, James O. Lloyd-Smith, Peggy Fong.

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
