## [Decision Letter · Decision Letter 0]

PCOMPBIOL-D-24-01560

Herbivore community function shapes resilience and bistability of coral reefs

PLOS Computational Biology

Dear Dr. Sura,

Thank you for submitting your manuscript to PLOS Computational Biology. After careful consideration, we feel that it has merit but does not fully meet PLOS Computational Biology's publication criteria as it currently stands. Therefore, we invite you to submit a revised version of the manuscript that addresses the points raised during the review process.

Please submit your revised manuscript within 60 days Apr 12 2025 11:59PM. If you will need more time than this to complete your revisions, please reply to this message or contact the journal office at ploscompbiol@plos.org. Please include the following items when submitting your revised manuscript:

We look forward to receiving your revised manuscript.

Kind regards,

Youhua Chen

Academic Editor

PLOS Computational Biology

Natalia Komarova

Section Editor

PLOS Computational Biology

**Additional Editor Comments :**

Both reviewers are positive about the conclusions and findings of the paper. However, I also agree with the second reviewer, detailed description should be given for the model presented in the paper, parameter used and the associated meanings, structure of the fish dynamics. I believe substantial work is needed.

**Journal Requirements:**

1) Please upload all main figures as separate Figure files in .tif or .eps format. For more information about how to convert and format your figure files please see our guidelines: 

2) We have noticed that you have uploaded Supporting Information files, but you have not included a list of legends. Please add a full list of legends for your Supporting Information files after the references list.

**Reviewers' comments:**

Reviewer's Responses to Questions

Reviewer #1: Summary

This paper uses a dynamical systems model with benthic (coral, macroalgae, turf) and herbivorous fish (browsers, grazers, generalists) components to explore the impact of herbivore community composition and fishing on the competitive outcomes between corals and algae. They find that a system comprised of all three herbivore types represents the most resilient reef in terms of maintaining and recovering a high coral cover state. Browser-dominated communities that target macroalgae are more resilient than grazer-dominated communities that target turf.

Major Comments

This is a well-written scientific paper and a nice contribution to the literature. The authors begin with a compelling introduction that clearly articulates the research questions and knowledge gaps they aim to address. The analytical approach is thoughtfully designed and clearly explained throughout the manuscript.The discussion section is particularly strong, opening with an effective summary of the broad findings and maintaining clear, crisp writing throughout in terms of both messaging and scientific justification.

The authors handle their relationship to related work well, particularly in their treatment of Cook et al.’s similar study. They appropriately acknowledge both the similarities and differences between the studies, noting that while approaches varied, the primary findings align - which serves to highlight the robustness of their results. Importantly, they recognize that the differences between studies help identify key model assumptions that warrant further empirical investigation.

Their findings offer implications for conservation and management, particularly regarding how the herbivore community influences system reversibility. The discussion excels at contextualizing their findings within recent empirical work and the broader literature while clearly outlining implications for both future research directions and management strategies.

I really enjoyed this paper and only have some minor comments and suggestions.

Minor Comments

A note about presenting parameter values: Default parameter values are typically presented in at least 1 figure caption and/or a table in the main text. Since you have an excellent detailed table in the supplement, I suggest simply listing parameter values in Fig 2 and then referring to Fig 2 caption while noting any deviations in the subsequent figures.

L82-83: I suggest providing mechanisms for how algal turfs and macroalgae facilitate and prevent coral recovery, respectively

L172-174: What about browser-dominated communities?

Fig 3: Are the final coral covers different across the coral-dominated region? Because if so, I would modify the color palette to make it more obvious

Fig 5: I suggest more clearly indicating bistability. Maybe dashed vertical lines to show the bifurcation fold? Also, I suggest adding arrows along the lines (for example, an arrow pointing down along the red forward line) to highlight directionality

L612: Did you mean to have the “C vs. C” in parentheses?

Reviewer #2: I quite liked this paper. It sits in a line of modeling trying to understand resilience of coral dominated states and to explain transitions from coral to macroalgal dominance. The paper is well grounded in the existing literature but pushes it forward with a focus on the different kinds of herbivorous fish.

The paper is quite similar to Cook at al. 2024 but has some differences in the model structure and in the questions being asked that I think are sufficient to make this a valuable contribution. Indeed, the second paragraph of the discussion reveals that this work was mostly done before the Cook paper was published and thus this paper finds itself accidently very close to published work. While I appreciate the honestly, and think it is very interesting to compare the two sets of results, I feel like the discussion is maybe a little too centered on comparison with the Cook study.

-----------

I have two substantive comments that require some attention from the authors and then a bunch of more minor suggestions.

My first substantive concern is with parameterization. I like that the authors stated some guiding principles in terms of the biological assumptions they are trying to meet. I went to the supplement to try to figure out how they got from those guiding principles to specific numbers and was disappointed not to find more details on these decisions.

So for example they say:

4. Mortality due to herbivory is greater for turf than macroalgae (gT > gM).

And I like that as a concept but then I get down to table S2 and they've just cut the mortality rate in half for macroalgae. And… maybe? Why not 25% why not 10%? I would like a little more in the supplement saying what constrained each of these choices. And which of these are “we just chose a smaller number” and which have a little more thought. I am not going to reject “we just chose a smaller number,” sometimes that is the best you can do, but it would be good to write down the thought process.

Arising from that, I am wondering which of their qualitative results are dependent on some of these poorly constrained decisions about parameters. I want to be careful here, because I don’t want to hold this paper to a higher standard than others (including my own) that often have to guess some plausible parameters. And I do not want to demand some big sensitivity analysis.

But I look at the results in the left column of figure 2 and the authors’ conclusion that “Reef recovery also occurs when the initial abundances of grazers versus browsers are more similar” and I wonder how much this depends on……

well for example principle 5 says:

“Macroalgae exert a stronger competitive effect on coral than turf (αM > αT).“

Which I accept. But that then gets converted into an alphaT of 0.25 and an alphaM of 0.5. But could alphaT not reasonably be set to zero? Lots of similar models do not even treat turf as preempting space from coral overgrowth, so other models have an effective competitive ability of turf even weaker than setting alphaT to zero. If alphaT was set to 0 (or 0.025 or some other very small number), would you still conclude that coral does best with a mix of grazers and browsers? Or would you find that really browsers are the key constraint?

Or if the gaps between GT and GM were greater (also totally plausible) how would that change qualitative results?

Not quite sure what to ask for. Again, a sensitivity analysis for all the parameters seems like overkill. But I would like to invite the authors to add some more supplementary material showing how the qualitative results hold up under some different choices for key parameters.

My second substantive concern is with the structure of the fish dynamics. On line 155 and other places the results are framed as selective fishing or protecting some functional groups from fishing. But I find this very odd given that fishing appears to be evenly applied to all groups. If the underlying question is about fishing why not create the different ratios of functional groups by reducing (or increasing) fishing on one group or another.

And then I started looking carefully at how the differences in the fish communities are set up and I realized that these groups are not individually regulated. So they have a shared carrying capacity, and whatever the ratio of groups at the start of the simulation seems to be preserved throughout. Indeed, this seems to require absolute symmetry in parameter values. If one functional group grows a tiny bit slower or dies a tiny bit faster, it will be rapidly shut out of the system (going to zero given the lack of external recruitment). This is partly acknowledged around line 515 (the equivalence of vital rates), but the text acts like that assumption could be relaxed, while I think it cannot be relaxed without building a more robust method of population regulation into the model, one that gives each group some growth rate advantage over the others when it is rare.

I think this might be fine for the results, which are mostly about how benthic dynamics are affected by the fish functional groups that happen to exist. But it represents a pretty serious fragility in model structure. And I think it means the authors ought to be very cautious about any claims they are making about fish dynamics or about the effect of fishing on this system.

More minor notes

I am somewhat thrown by the amount of material that is in each figure caption. But this is a style point and not wrong

Figure 1 seems like 2 figures stuck together (is this because there is a limit to how many can be in main text?) if not maybe separate the diagram from the sample results

I wish there were more sample results so I could get some intuition about the model, maybe in a supplement. That might just be me.

In figure 2 I wonder if all these triangles could really be represented as ratios. Why not just show the equilibrium ratio of grazers to browsers on the x and have a line for coral outcomes under each fishing on the y axis? The figure is fine, but a lot of ink and I suspect starting with .6 grazer and .3 browser is the same as starting at .4 grazer and .2 browser.

On line 329-33 again there is discussion of the results as simulating overfishing, but that does not seem to be what the model does. I almost wonder If there were overfishing analyses that did not get included in this draft?

365 same problem. The model did not examine targeted fishing practices.

The paragraph starting on line 384 covers population regulation. I don’t think it quite characterizes Cook correctly. I looked back at that paper and they have separate population regulation for each species (each has it’s own carrying capacity). And those carrying capacities do not depend on food resources as is implied here. Growth rates do depend on food, so I guess maybe it works out that the fish equilibria depend on algal community. But it is a pretty different form than is used here, and it might be worth discussion the implications of each further, or at least clarifying how regulation worked in the cook model.

I found the discussion starting on line 431 very interesting. I agree that the results here are different from cook because the model here has more ways of creating positive feedbacks (and in particular, once you lose coral fish can’t come back). [That is just me editorializing not a request to add any text.]

The authors frame the results in terms of diversity in the conclusions and at the end of the abstract. I don’t love it because there is so little diversity represented here, but I don’t have a real problem with it.

On line 596 maybe rephrase the way fishing is described? Fishing is just a consonant constant per capita loss rate. The phrase here makes it seem like fishing is somehow dependent on abundance which is not what I see in the equations

Line 611 could use units presumably per year

I really like the right side of figure 5, I don’t think the left side adds much? This is sort of a question for the editor whether space matters.

**Have the authors made all data and (if applicable) computational code underlying the findings in their manuscript fully available?**

Reviewer #1: Yes

Reviewer #2: Yes

PLOS authors have the option to publish the peer review history of their article (what does this mean? ). If published, this will include your full peer review and any attached files.

**Do you want your identity to be public for this peer review?** For information about this choice, including consent withdrawal, please see our Privacy Policy .

Reviewer #1: No

Reviewer #2: No

**Figure resubmission:**
---

## [Decision Letter · Decision Letter 1]

Dear Dr. Sura,

We are pleased to inform you that your manuscript 'Herbivore community function shapes resilience and bistability of coral reefs' has been provisionally accepted for publication in PLOS Computational Biology.

Best regards,

Youhua Chen

Academic Editor

PLOS Computational Biology

Natalia Komarova

Section Editor

PLOS Computational Biology

The reviewers have positive feedbacks on your paper, congratulations! Moreover, it is possible, try to integrate the comments of the second reviewer on the sensitivity analysis in your final draft for final publications.

Reviewer's Responses to Questions

**Comments to the Authors:**

Reviewer #1: The authors have thoroughly addressed all of my concerns which were mostly points of clarification and some figure suggestions. I also appreciated their thoughtful responses to the other reviewer, and I think the addition of the sensitivity analyses in the supplement are a helpful addition.

Reviewer #2: I still really like this paper.

The edits in response to my review comments are well done. My important comments are accounted for but the authors push back in places which is fine.

I will say I was disappointed by the sensitivity analysis. A huge pile of graphs were added to the supplement, but it is very hard for me to know what to do with these. In my review I said: “I would like to invite the authors to add some more supplementary material showing how the qualitative results hold up under some different choices for key parameters.”

And…there are a lot of graphs here, so maybe I could figure out those answers?

But for example figure 6 in the main text argues that the alternative states are very sensitive to the mix of functional groups – the abstract describes a “loss of alternative stable states for browser-dominated communities, or continued presence of alternative stable states in grazer-dominated communities even when fishing pressure is removed entirely.”

I would have thought a sensitivity analysis would show versions of figure 6 under slightly different assumptions about various parameters, assessing whether this qualitative patterns holds for different reasonable representations of benthic dynamics. Figure S10 seems to just show the same parameters but for increasing dominance of browsers? How robust is this pattern to all the other parameter decisions made

(as an aside B0 is introduced here in the caption for figure S10 but appears nowhere else, though its meaning is clear from context)

In my initial review I said I don’t want to ask for a full sensitivity analysis, and so I am going to stand by that and say this is all fine. But I do want to register that this added material does not really answer my question about how robust the qualitative results are.

**Have the authors made all data and (if applicable) computational code underlying the findings in their manuscript fully available?**

Reviewer #1: Yes

Reviewer #2: Yes

PLOS authors have the option to publish the peer review history of their article (what does this mean? ). If published, this will include your full peer review and any attached files.

**Do you want your identity to be public for this peer review?** For information about this choice, including consent withdrawal, please see our Privacy Policy .

Reviewer #1: No

Reviewer #2: No

---

## [Editor Report · Acceptance letter]

PCOMPBIOL-D-24-01560R1

Herbivore community function shapes resilience and bistability of coral reefs

Dear Dr Sura,

I am pleased to inform you that your manuscript has been formally accepted for publication in PLOS Computational Biology. Your manuscript is now with our production department and you will be notified of the publication date in due course.

With kind regards,

Zsofia Freund
